# TEST-TIME ITERATIVE ERROR CORRECTION FOR EFFICIENT DIFFUSION MODELS

**Yunshan Zhong[1], Weiqi Yan[2], Yuxin Zhang[2]***
[1]School of Computer Science and Technology, Hainan University
[2]MAC Lab, Department of Artificial Intelligence, School of Informatics, Xiamen University
`yszhong01@gmail.com, weiqi_yan@outlook.com, yuxinzhang@stu.xmu.edu.cn`

## ABSTRACT

With the growing demand for high-quality image generation on resource-constrained devices, efficient diffusion models have received increasing attention. However, such models suffer from approximation errors introduced by efficiency techniques, which significantly degrade generation quality. Once deployed, these errors are difficult to correct, as modifying the model is typically infeasible in deployment environments. Through an analysis of error propagation across diffusion timesteps, we reveal that these approximation errors can accumulate exponentially, severely impairing output quality. Motivated by this insight, we propose Iterative Error Correction (IEC), a novel test-time method that mitigates inference-time errors by iteratively refining the model's output. IEC is theoretically proven to reduce error propagation from exponential to linear growth, without requiring any retraining or architectural changes. IEC can seamlessly integrate into the inference process of existing diffusion models, enabling a flexible trade-off between performance and efficiency. Extensive experiments show that IEC consistently improves generation quality across various datasets, efficiency techniques, and model architectures, establishing it as a practical and generalizable solution for test-time enhancement of efficient diffusion models. The code is available at `https://github.com/zysxmu/IEC`.

## 1 INTRODUCTION

Diffusion models (Ho et al., 2020; Rombach et al., 2022; Nichol & Dhariwal, 2021) have achieved state-of-the-art generative performance across a wide range of tasks, including image synthesis (Song et al., 2020b; Choi et al., 2021; Saharia et al., 2022a; Tumanyan et al., 2023; Li et al., 2022; Saharia et al., 2022c; Gao et al., 2023; Avrahami et al., 2022; Kawar et al., 2023; Meng et al., 2021), text-to-image generation (Nichol et al., 2021; Ramesh et al., 2022; Saharia et al., 2022b; Zhang et al., 2023b), text-to-3D generation (Lin et al., 2023; Luo & Hu, 2021; Poole et al., 2022), video generation (Mei & Patel, 2023), and audio generation (Huang et al., 2022; Zhang et al., 2023a). However, diffusion models typically require hundreds of iterative denoising steps and contain billions of parameters, resulting in high computational costs that hinder deployment in resource-constrained environments. To address this limitation, extensive efficiency techniques have been devoted to developing efficient model techniques for diffusion models (Song et al., 2020a; Zhang et al., 2022; Zheng et al., 2023; Liu et al., 2025b; Shang et al., 2023; Li et al., 2023; 2024c).

Among these techniques, network quantization (Shang et al., 2023; Li et al., 2023; 2024c; Zheng et al., 2024) and feature caching (Chen et al., 2024b; Zou et al., 2024; Ma et al., 2024b) have emerged as two promising approaches. The former reduces data precision to low-bit representations to simultaneously shrink model size and accelerate inference, while the latter caches intermediate features to eliminate redundant computations across diffusion timesteps. Although these methods effectively reduce overhead, they inevitably suffer from approximation errors between the efficient model and the original counterpart, which degrade the generation quality of the model. Prior studies have attempted to mitigate such errors through specialized mechanisms. For example, timestep-wise quantization parameters (Wang et al., 2023; Liu et al., 2024b) have been introduced to capture the dynamics of

---

*Corresponding Author: yuxinzhang@stu.xmu.edu.cn

time-varying activations, while non-uniform caching strategies (Ma et al., 2024b) exploit similarity patterns between adjacent timesteps to improve performance. Despite their effectiveness, these methods remain pre-deployment solutions that require the ability to re-execute the model-efficiency pipeline and the original model.

In practice, such requirements often do not hold once a model has been deployed in edge or production environments. On the one hand, reapplying the model-efficiency pipeline and redeploying the model incurs significant engineering overhead, making it impractical in many cases. Also, deployed models are typically immutable due to storage limitations, deployment policies, or system design constraints. On the other hand, after being deployed, the original model is often irretrievable, making re-executing the model-efficiency pipeline unfeasible. For instance, a quantized model may no longer retain its original high-precision weights, making re-quantization infeasible. Inspired by recent advances in test-time scaling (Snell et al., 2024; Lightman et al., 2023; Muennighoff et al., 2025; Ma et al., 2025; Singhal et al., 2025; Prabhudesai et al., 2023; Xiao & Snoek, 2024), where model behavior is adjusted at inference time without retraining, we ask: *Is it possible to improve the performance of an already deployed diffusion model without repeating the model-efficiency pipeline?*

To answer this question, we begin by analyzing how errors propagate through diffusion timesteps and reveal that they accumulate exponentially, which significantly degrades the final generation quality. Motivated by this finding, we propose Iterative Error Correction (IEC), a novel test-time method that mitigates these errors by iteratively refining the model's output. Theoretically, we show that IEC reduces error accumulation from exponential to linear growth. IEC is a plug-and-play method that operates entirely at test-time. It requires no re-running the model-efficiency pipeline, no fine-tuning weights, and no changes in model architecture, making it compatible with deployed diffusion models. While IEC introduces additional computational overhead, it is highly flexible and can be selectively applied to a subset of all diffusion timesteps. Applying IEC to more timesteps yields greater quality improvements, while using fewer timesteps reduces computational overhead. Skipping IEC entirely preserves the model's original performance. This flexibility provides users with fine-grained control over the trade-off between efficiency and generation quality, depending on resource constraints and application needs. By enhancing the performance of efficient diffusion models in deployment, IEC preserves the compatibility and reusability of these models, making it a practical and generalizable solution for real-world applications.

## 2 RELATED WORK

### 2.1 DIFFUSION MODEL

Diffusion models (Ho et al., 2020; Rombach et al., 2022; Karras et al., 2022; 2024; Peebles & Xie, 2023) consist of *forward process* and *reverse process*. In the *forward process*, given input data distribution $x_0 \sim q(x)$, diffusion models add a series of Gaussian noise to $x_0$ to resulting in a sequence of noisy samples $x_t, 0 \le t \le T$:

$$q(x_{1:T}|x_0) = \prod_{t=1}^{T} q(x_t|x_{t-1}),$$
$$q(x_t|x_{t-1}) = \mathcal{N}(x_t; \sqrt{\alpha_t}x_{t-1}, \beta_t \mathbf{I}), \tag{1}$$

where $\alpha_t = 1 - \beta_t$, $\beta_t$ is $t$-based parameters. In the *reverse process*, given randomly sampled Gaussian noise $x_T \sim \mathcal{N}(\mathbf{0}, \mathbf{I})$, diffusion models progressively generate images by:

$$p_\theta(x_{t-1}|x_t) = \mathcal{N}(x_{t-1}; \hat{\mu}_{\theta,t}(x_t), \hat{\beta}_t \mathbf{I}). \tag{2}$$

For DDIM (Song et al., 2020a), the $\hat{\beta}_t = 0$ and $\hat{\mu}_{\theta,t}$ is defined by:

$$x_{t-1} = \sqrt{\alpha_{t-1}} \frac{x_t - \sqrt{1-\alpha_t}\epsilon_\theta(x_t, t)}{\sqrt{\alpha_t}} + \sqrt{1-\alpha_{t-1}}\epsilon_\theta(x_t, t). \tag{3}$$

### 2.2 EFFICIENT DIFFUSION

Significant efforts have been devoted to developing efficient diffusion models, which can be broadly categorized into two types: temporal efficiency and structural efficiency (Liu et al., 2025b). Temporal

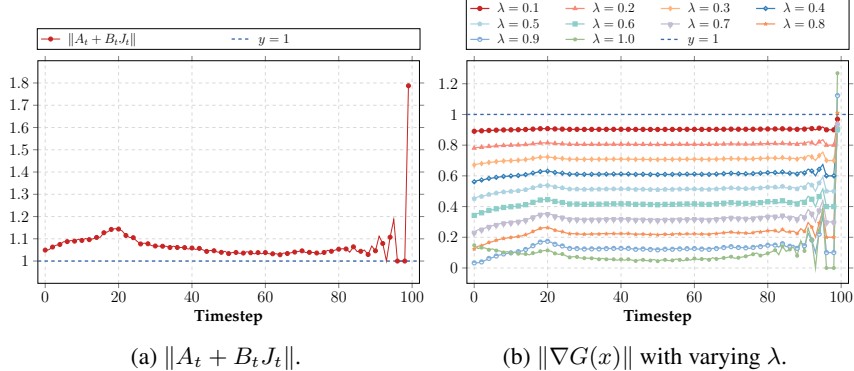

Figure 1: Empirical results of (a) $\|A_t + B_t J_t\|$; (b) $\|\nabla G(x)\|$ under varying $\lambda$. Reported values are averaged over 100 sample generations using a DDIM pretrained on CIFAR-10.

efficiency methods focus on reducing the number of sampling timesteps. For instance, methods such as DDIM (Song et al., 2020a) and GDDIM (Zhang et al., 2022) modify the denoising equations to enable fewer sampling steps, while others (Zheng et al., 2023; Shih et al., 2023) accelerate inference by performing multiple denoising timesteps in parallel. DistriFusion (Li et al., 2024a) further improves efficiency by dividing input patches across multiple GPUs. Additionally, fast solvers for diffusion (Liu et al., 2022; Dockhorn et al., 2022; Yin et al., 2024b;a; Lu et al., 2022a;b) have been proposed to reduce the computational overhead associated with sampling. Structural efficiency methods, on the other hand, aim to reduce model complexity and are complementary to temporal efficiency techniques, making them increasingly popular. For example, network quantization has emerged as a key approach for efficient diffusion model (Li et al., 2024c; Zheng et al., 2024; Yang et al., 2024; Deng et al., 2025; Zhao et al., 2024; Huang et al., 2024a; Chen et al., 2024a; Dong & Zhang, 2025). Quantization-aware training (QAT)(Li et al., 2024c; Zheng et al., 2024) restores model performance at low bit-widths but incurs significant training overhead. In contrast, post-training quantization (PTQ)(Shang et al., 2023; Li et al., 2023; Wu et al., 2024a) requires minimal computational resources and small calibration datasets, making it more suitable for scenarios with limited resources. PTQ4DM (Shang et al., 2023) and Q-Diffusion (Li et al., 2023) introduced time-aware calibration to realize quantized diffusion models. Subsequent works have refined these methods by addressing quantization error, feature re-balancing, calibration data collection, model reconstruction, mixed-precision strategies, and so on (He et al., 2024; Huang et al., 2024b; Wang et al., 2024a;b; Feng et al., 2025; Liu et al., 2024c; Sun et al., 2024; Wu et al., 2024b). Moreover, techniques such as LoRA-based optimization (Hu et al., 2021) have been applied to further enhance quantized diffusion models (Li et al., 2024b; He et al., 2023). In addition to quantization, feature caching has also gained attention (Tang et al., 2024; Chen et al., 2024b; Zou et al., 2024; Ma et al., 2024b; Wimbauer et al., 2024; Liu et al., 2025a; 2024a; Ma et al., 2024a). These methods aim to reduce inference time by reusing pre-computed features across diffusion steps. Recent studies, such as CacheQuant (Liu et al., 2025b), combine caching with quantization to achieve even greater efficiency. Other structural efficiency techniques include token pruning (Fang et al., 2024) and sparsity (Fan et al., 2025). In this work, we focus on mitigating the errors introduced by structural efficiency methods.

## 3 METHOD

### 3.1 ANALYSIS OF ERROR ACCUMULATION IN EFFICIENT DIFFUSION MODELS

In this subsection, we present a theoretical analysis of error propagation and accumulation across diffusion timesteps, using the deterministic DDIM sampling procedure as an example.

**Preliminaries.** We consider the deterministic DDIM sampling process defined as:

$$x_{t-1} = \sqrt{\alpha_{t-1}}\frac{x_t - \sqrt{1-\alpha_t}\epsilon_\theta(x_t, t)}{\sqrt{\alpha_t}} + \sqrt{1-\alpha_{t-1}}\epsilon_\theta(x_t, t). \tag{4}$$

For brevity of notation, we define the following coefficients:

$$A_t = \frac{\sqrt{\alpha_{t-1}}}{\sqrt{\alpha_t}}, \quad B_t = \sqrt{1 - \alpha_{t-1}} - \sqrt{\alpha_{t-1}}\frac{\sqrt{1 - \alpha_t}}{\sqrt{\alpha_t}}, \tag{5}$$

thereby simplifying the rule to:

$$x_{t-1} = A_t x_t + B_t \epsilon_\theta(x_t, t). \tag{6}$$

**Modeling Error.** In efficient diffusion model methods, such as quantized or cached models, two primary types of errors are introduced at each timestep. First, due to error propagation from the previous timestep, there exists a discrepancy $\delta_t$ between the perturbed input $\tilde{x}_t$ and the ideal input $x_t$, defined as $\tilde{x}_t = x_t + \delta_t$. Second, perturbations from network quantization or feature caching introduce an error $\epsilon_\theta^\delta$ in the model's prediction. Specifically, the perturbed prediction $\tilde{\epsilon}_\theta(x_t, t)$ differs from the ideal case $\epsilon_\theta(x_t, t)$ is defined as $\tilde{\epsilon}_\theta(x_t, t) = \epsilon_\theta(x_t, t) + \epsilon_\theta^\delta$.

Incorporating these errors, the perturbed update rule at timestep $t$ becomes:

$$\begin{aligned}
\tilde{x}_{t-1} &= A_t \tilde{x}_t + B_t \tilde{\epsilon}_\theta(\tilde{x}_t, t) \\
&= A_t(x_t + \delta_t) + B_t \tilde{\epsilon}_\theta(x_t + \delta_t, t) \\
&\approx A_t(x_t + \delta_t) + B_t \left( \tilde{\epsilon}_\theta(x_t, t) + J_t \delta_t \right) \\
&= A_t(x_t + \delta_t) + B_t \left( \epsilon_\theta(x_t, t) + \epsilon_\theta^\delta + J_t \delta_t \right),
\end{aligned} \tag{7}$$

where the approximation uses the first-order Taylor expansion, and $J_t = \frac{\partial \tilde{\epsilon}_\theta(x,t)}{\partial x}\big|_{x=x_t}$ represents the Jacobian of the model output with respect to its input.

**Error Propagation.** By subtracting the ideal update in Eq. 6 from the perturbed update in Eq. 7, we derive the recursive relation for error propagation:

$$\begin{aligned}
\delta_{t-1} &= \tilde{x}_{t-1} - x_{t-1} \\
&\approx A_t \delta_t + B_t(\epsilon_\theta^\delta + J_t \delta_t) \\
&= (A_t + B_t J_t)\delta_t + B_t \epsilon_\theta^\delta.
\end{aligned} \tag{8}$$

Recursively expanding Eq. 8 from timestep $T$ to 0, and assuming the initial error $\delta_T = 0$ (since $x_T$ is drawn from an ideal Gaussian prior), the accumulated error at $t = 0$ is given by:

$$\delta_0 = \sum_{i=1}^{T} \left( \prod_{j=i+1}^{T} (A_j + B_j J_j) \right) (B_i \epsilon_\theta^\delta). \tag{9}$$

Eq. 9 reveals that the final error $\delta_0$ is a weighted sum of per-timestep prediction errors, where each contribution is scaled by the product of the matrices $(A_j + B_j J_j)$ from timestep $i + 1$ to $T$.

**Analysis of Error Growth.** To understand whether the propagated error amplifies or decays over time, we analyze the spectral norm $\|A_t + B_t J_t\|$, which quantifies the maximum amplification of the linear transformation at each timestep. Specifically, if $\|A_t + B_t J_t\| > 1$, the corresponding error component is amplified, and the error can grow exponentially if such amplification persists, leading to instability. In contrast, if $\|A_t + B_t J_t\| < 1$ for all $t$, the propagated error decays over time, indicating a relatively stable and robust sampling process. Empirical observations, as shown in Fig. 1a, demonstrate that $\|A_t + B_t J_t\|$ consistently exceeds 1 across timesteps, suggesting that the errors introduced by efficient diffusion methods tend to accumulate exponentially, posing a challenge in stability.

As shown in Eq. 9, the key to reducing error lies in breaking the exponential growth trend during the accumulation process. In the following subsection, we propose a novel and effective method that transforms the error growth from exponential to linear.

## 3.2 ITERATIVE ERROR CORRECTION

To mitigate error accumulation, we introduce Iterative Error Correction (IEC), a theoretically motivated, plug-and-play method designed to reduce error accumulation from exponential to linear growth at test-time. The core idea of IEC is to introduce correction steps within diffusion timesteps. Specifically, at a timestep $t-1$, an initial estimate $x_{t-1}^{(0)}$ is computed using the standard DDIM update defined in Eq. 6: $x_{t-1}^{(0)} = A_t x_t + B_t \epsilon_\theta(x_t, t)$. We then iteratively refine this estimate by repeatedly applying the following update rule:

$$x_{t-1}^{(k+1)} = x_{t-1}^{(k)} + \lambda \left( A_t x_t + B_t \epsilon_\theta(x_{t-1}^{(k)}, t) - x_{t-1}^{(k)} \right),$$
$$k = 0, 1, 2, \ldots, \tag{10}$$

where $\lambda$ is a tunable hyperparameter. The iteration proceeds until the difference $\|x_{t-1}^{(k+1)} - x_{t-1}^{(k)}\|$ falls below a predefined threshold or the maximum number of iterations is reached, yielding the final estimate $x_{t-1}^*$.

**Convergence Analysis.** We provide a mathematical justification for the convergence of IEC using fixed-point theory. Eq. 10 can be reformulated as:

$$x_{t-1}^{(k+1)} = (1-\lambda)x_{t-1}^{(k)} + \lambda \left( A_t x_t + B_t \epsilon_\theta(x_{t-1}^{(k)}, t) \right). \tag{11}$$

This allows us to define the mapping $G(x)$ as:

$$G(x) = (1-\lambda)x + \lambda \left( A_t x_t + B_t \epsilon_\theta(x, t) \right). \tag{12}$$

The iterative procedure in Eq. 10 is thus equivalent to applying fixed-point iteration to solve:

$$x_{t-1}^* = G(x_{t-1}^*). \tag{13}$$

According to Banach's fixed-point theorem (Banach, 1922), the iterative procedure converges to a unique fixed point $x_{t-1}^*$ if the mapping $G(x)$ is a contraction mapping, *i.e.*, there exists a Lipschitz constant $0 < L < 1$ such that:

$$\|G(x) - G(y)\| \leq L\|x - y\|, \forall x, y. \tag{14}$$

To estimate the Lipschitz constant $L$, we compute the Jacobian of $G(x)$:

$$\nabla G(x) = (1-\lambda)I + \lambda B_t J_t, \tag{15}$$

where $J_t$ is the Jacobian matrix. Then, the Lipschitz constant $L$ is given by:

$$L = \|\nabla G(x)\| = \|(1-\lambda)I + \lambda B_t J_t\|. \tag{16}$$

To guarantee convergence, it is necessary to satisfy $0 < L < 1$. Since $B_t < 0$, an appropriately chosen positive $\lambda$ reduces the Lipschitz constant $L$ via the term $\lambda B_t J_t$, ensuring $L < 1$. Empirically, as shown in Fig. 1b, we observe that setting $\lambda$ within the range [0.1, 0.7] consistently ensures that $\|\nabla G(x)\| < 1$ across all timesteps. In this paper, we set $\lambda$ to 0.5 as a practical choice based on these observations. Moreover, since $G(x)$ is continuously differentiable, Eq. 14 holds by the Mean Value Inequality for vector-valued functions (Munkres, 2018), confirming that $G$ is a contraction mapping and IEC can converge to the fixed-point solution.

**Error Accumulation in IEC.** The proposed IEC effectively suppresses error accumulation. To demonstrate this, we define the error at iteration $k+1$ of IEC as:

$$\delta_{t-1}^{(k+1)} = x_{t-1}^{(k+1)} - x_{t-1} = G(x_{t-1}^{(k)}) - x_{t-1}$$
$$= \underbrace{G(x_{t-1} + \delta_{t-1}^{(k)}) - G(x_{t-1})}_{\text{first term}} + \underbrace{G(x_{t-1}) - x_{t-1}}_{\text{second term}}, \tag{17}$$

where $\delta_{t-1}^{(k)} = x_{t-1}^{(k)} - x_{t-1}$ is the accumulated error at iteration $k$. The first term of Eq. 17 can be approximated by $G(x_{t-1} + \delta_{t-1}^{(k)}) - G(x_{t-1}) \approx \nabla G \cdot \delta_{t-1}^{(k)}$.

By considering the noisy input and noisy model prediction, the mapping $G(x)$ is defined as:

$$G(x) = (1 - \lambda)x + \lambda(A_t \tilde{x}_t + B_t \tilde{\epsilon}_\theta(x, t)), \tag{18}$$

where $\tilde{x}_t = x_t + \delta_t$ and $\tilde{\epsilon}_\theta(x, t) = \epsilon_\theta(x, t) + \epsilon_\theta^\delta$ represent the noisy input and noisy model prediction, respectively. Consequently, the second term in Eq. 17 can be approximated by:

$$G(x_{t-1}) - x_{t-1} =$$
$$(1 - \lambda)x_{t-1} + \lambda(A_t(x_t + \delta_t) + B_t(\epsilon_\theta(x_{t-1}, t) + \epsilon_\theta^\delta)) - x_{t-1}$$
$$= \lambda(A_t x_t + B_t \epsilon_\theta(x_{t-1}, t) - x_{t-1}) + \lambda(A_t \delta_t + B_t \epsilon_\theta^\delta)$$
$$= \lambda B_t(\epsilon_\theta(x_{t-1}, t) - \epsilon_\theta(x_t, t)) + \lambda(A_t \delta_t + B_t \epsilon_\theta^\delta),$$

where the last equality uses the relationship $x_{t-1} = A_t x_t + B_t \epsilon_\theta(x_t, t)$. Taking the norm of $\delta_{t-1}^{(k+1)}$ and applying the triangle inequality, we obtain:

$$\|\delta_{t-1}^{(k+1)}\| \leq \|\nabla G(x_{t-1}) \cdot \delta_{t-1}^{(k)}\| +$$
$$\lambda(\|B_t(\epsilon_\theta(x_{t-1}, t) - \epsilon_\theta(x_t, t))\| + \|A_t \delta_t + B_t \epsilon_\theta^\delta\|)$$
$$\leq L\|\delta_{t-1}^{(k)}\| + C,$$

where $L = \|\nabla G(x_{t-1})\|$ is the Lipschitz constant, and $C = \lambda\left(\|B_t(\epsilon_\theta(x_{t-1}, t) - \epsilon_\theta(x_t, t))\| + \|A_t \delta_t + B_t \epsilon_\theta^\delta\|\right)$ is a bounded constant independent of $\delta_{t-1}^{(k)}$. Recursively applying Eq. 19 yields:

$$\|\delta_{t-1}^{(k)}\| \leq L^k \|\delta_{t-1}^{(0)}\| + C\frac{1 - L^k}{1 - L}. \tag{19}$$

As $k \to \infty$, $L^k \to 0$, and the error converges to:

$$\|\delta_{t-1}^{(\infty)}\| \leq \frac{C}{1 - L}. \tag{20}$$

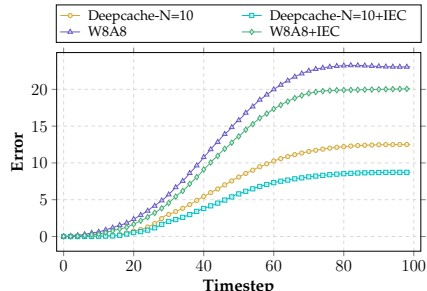

Figure 2: Error comparison across timesteps.

This result demonstrates that IEC effectively suppresses error accumulation by ensuring that the propagated error at each timestep is bounded by $\frac{C}{1-L}$. Crucially, since IEC eliminates dependency on errors from previous timesteps, the total accumulated error after $T$ timesteps grows only linearly: $\delta_0^{\text{IEC}} = \sum_{j=1}^T \delta_j^x$, where each $\delta_j^x$ is independently bounded. Thus, IEC can prevent exponential error amplification and achieve linear error propagation in theory. In practice, we set the maximum iteration K to 1 and the threshold $\tau$ to 1e-5. As shown in Fig. 2, IEC effectively reduces errors across timesteps, demonstrating its effectiveness in a real case.

## 4 EXPERIMENT

### 4.1 EXPERIMENT SETTINGS

**Models, Baselines, Datasets, Metrics, and Implementation Details.** All experiments are conducted using PyTorch on an NVIDIA 3090 GPU. To evaluate the effectiveness of IEC, we apply it to various diffusion models, including DDPM, LDM, and Stable Diffusion (Song et al., 2020a; Rombach et al., 2022)[1], as well as efficiency techniques such as timestep-wise network quantization (Liu et al., 2024b), Deepcache (Ma et al., 2024b), and CacheQuant (Liu et al., 2025b), a hybrid technique combining quantization and feature caching. For model quantization, we adopt channel-wise quantization for weights and layer-wise quantization for activations, considering both W4A8 and W8A8 cases. For W4A8, local reconstruction is performed to enhance performance. For feature caching, following

---

[1]We adopt the CompVis codebase and the official v1.4 checkpoint.

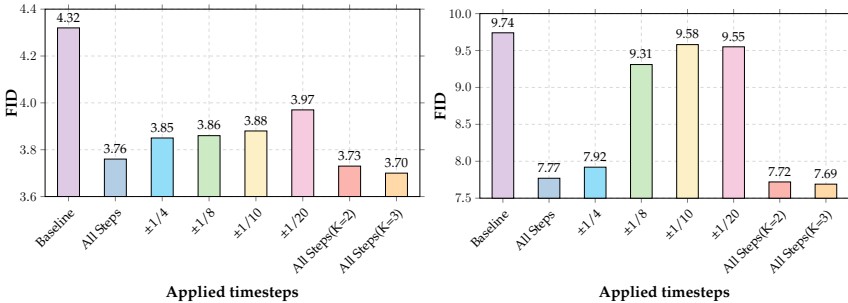

(a) Ablation study of applying IEC on W8A8 quantization.

(b) Ablation study of applying IEC on Deepcache (N = 10).

Figure 3: Ablation study of applying IEC on quantization and Deepcache. The baseline does not use IEC, while "All Steps" applies IEC to all timesteps. The "±A" indicates that IEC is applied to both the first and last A timesteps.

CacheQuant (Ma et al., 2024b), the cached blocks are set to last 3, 1, and 1 blocks for DDIM, LDM, and Stable Diffusion, respectively. We evaluate IEC on several widely used datasets, including CIFAR-10 (Krizhevsky et al., 2009), LSUN-Churchs (Yu et al., 2015), LSUN-Bedrooms (Yu et al., 2015), ImageNet (Deng et al., 2009), and MS-COCO (Lin et al., 2014). For Stable Diffusion, we generate 5,000 images using MS-COCO captions as prompts, following the protocols in (Ma et al., 2024b; Chen et al., 2024b; Wu et al., 2024a). For all other datasets, 50,000 images are generated to evaluate the quality of image synthesis. The evaluation metrics include FID (Heusel et al., 2017), Inception Score (IS) (Salimans et al., 2016), and CLIP Score (evaluated on ViT-g/14) (Hessel et al., 2021). For quantization-based methods, IEC is applied at every timestep. For DeepCache and CacheQuant, IEC is applied only to the non-cached timesteps, except for MS-COCO, where IEC is only applied at the first timestep. Note that IEC is applied solely to the efficient diffusion models and does not interfere with the baseline model efficiency pipelines.

Table 1: Results of combining IEC with timestep-wise quantization (Liu et al., 2024b) on CIFAR-10, LSUN-Churchs, and LSUN-Bedrooms datasets.

| CIFAR-10 $32 \times 32$ (T=100) | | LSUN-Churchs $256 \times 256$ (T=100) | | LSUN-Bedrooms $256 \times 256$ (T=100) | |
|---|---|---|---|---|---|
| W/A | FID $\downarrow$ | W/A | FID $\downarrow$ | W/A | FID $\downarrow$ |
| DDIM | 4.19 | LDM-8 | 3.99 | LDM-4 | 3.37 |
| W8A8 +IEC | 4.32 **3.76** | W8A8 +IEC | 3.57 **3.29** | W8A8 +IEC | 8.97 **7.78** |
| W4A8 +IEC | 6.82 **5.96** | W4A8 +IEC | 6.27 **6.10** | - - | - - |

## 4.2 ABLATION STUDY

Fig. 3 illustrates the impact of applying IEC to different subsets of timesteps. Specifically, we examine the effects of applying IEC only to the first and last timesteps, as these timesteps exhibit the largest values of $\|A_t + B_t J_t\|$ in Fig. 1a. As shown in Fig. 3a, for the W8A8 case, applying IEC to all timesteps achieves the best FID of 3.76. Notably, the performance gains remain significant even when IEC is applied to only a small number of timesteps. For instance, applying IEC to the first and last 1/10 ("±1/10") or 1/20 ("±1/20") of timesteps yields FID improvements of 0.44 and 0.35, respectively. These results suggest that partial application of IEC still provides meaningful benefits. A similar trend is observed in results on feature caching in Fig. 3b. Applying IEC to all timesteps improves the FID by 1.97, while employing it to the first and last 1/10 or 1/20 of timesteps leads to gains of 0.16 and 0.19, respectively. These findings demonstrate that IEC is effective and flexible, as it can be selectively applied to a subset of timesteps to balance performance improvements with

Table 2: Results of combining IEC with Deepcache (Ma et al., 2024b) on CIFAR-10, LSUN-Churchs, and LSUN-Bedrooms datasets.

| CIFAR 32 × 32 (T=100) | | LSUN-Churchs 256 × 256 (T=100) | | LSUN-Bedrooms 256 × 256 (T=100) | |
|---|---|---|---|---|---|
| Method | FID ↓ | Method | FID ↓ | Method | FID ↓ |
| DDIM | 4.19 | LDM-8 | 3.99 | LDM-4 | 3.37 |
| Deepcache-N=3 +IEC | 4.70 **3.96** | Deepcache-N=3 +IEC | 5.10 **4.73** | Deepcache-N=2 +IEC | 11.21 **7.99** |
| Deepcache-N=5 +IEC | 5.73 **4.83** | Deepcache-N=5 +IEC | 6.74 **6.00** | Deepcache-N=3 +IEC | 11.86 **8.16** |
| Deepcache-N=10 +IEC | 9.74 **7.77** | Deepcache-N=10 +IEC | 14.81 **13.17** | Deepcache-N=5 +IEC | 14.28 **9.20** |
| Deepcache-N=15 +IEC | 17.21 **14.58** | Deepcache-N=15 +IEC | 25.27 **22.42** | Deepcache-N=10 +IEC | 26.09 **16.91** |

computational cost[2]. Moreover, increasing the number of correction iterations (e.g., K = 2 or K = 3) leads to marginal improvements, indicating that IEC is already effective even with a single correction step.

Table 3: Results of combining IEC with CacheQuant (Liu et al., 2025b) (Denoted as CacheQ) on CIFAR-10, LSUN-Churchs, and LSUN-Bedrooms datasets.

| W/A | CIFAR-10 32 × 32 (T=100) | | LSUN-Churchs 256 × 256 (T=100) | | LSUN-Bedrooms 256 × 256 (T=100) | |
|---|---|---|---|---|---|---|
| | Method | FID ↓ | Method | FID ↓ | Method | FID ↓ |
| - | DDIM | 4.19 | LDM-8 | 3.99 | LDM-4 | 3.37 |
| 8/8 | CacheQ-N=3 +IEC | 4.61 **3.93** | CacheQ-N=3 +IEC | 3.66 **3.39** | CacheQ-N=2 +IEC | 8.85 **7.51** |
| | CacheQ-N=5 +IEC | 5.28 **5.09** | CacheQ-N=5 +IEC | 3.71 **3.24** | CacheQ-N=3 +IEC | 9.27 **7.33** |
| | CacheQ-N=10 +IEC | 8.19 **6.47** | CacheQ-N=10 +IEC | 5.54 **4.25** | CacheQ-N=5 +IEC | 10.29 **7.67** |
| | CacheQ-N=15 +IEC | 13.42 **10.77** | CacheQ-N=15 +IEC | 9.47 **6.90** | CacheQ-N=10 +IEC | 17.53 **11.07** |
| 4/8 | CacheQ-N=3 +IEC | 7.27 **6.42** | CacheQ-N=3 +IEC | 7.08 **6.85** | - | - |
| | CacheQ-N=5 +IEC | 8.15 **6.90** | CacheQ-N=5 +IEC | 7.24 **6.79** | - | - |
| | CacheQ-N=10 +IEC | 11.36 **10.69** | CacheQ-N=10 +IEC | 10.75 **9.69** | - | - |
| | CacheQ-N=15 +IEC | 16.76 **15.50** | CacheQ-N=15 +IEC | 13.28 **11.50** | - | - |

## 4.3 MAIN RESULTS

This subsection includes the quantitative results, while the visualizations are included in the appendix.

**Evaluation on IEC on Network Quantization.** Tab. 1 presents the quantization performance. For LSUN-Bedrooms, we report only W8A8 results, as W4A8 quantization leads to model collapse.

---

[2]In Sec. 4.4, we provide discussion about the time overhead of IEC.

Across all datasets, applying IEC consistently improves performance. For example, in the W4A8 case, IEC reduces the FID from 6.82 to 5.96 on CIFAR-10, and from 6.27 to 6.10 on LSUN-Churchs.

**Evaluation on IEC on Feature Caching.** The performance of IEC on DeepCache (Ma et al., 2024b) is shown in Tab. 2, demonstrating consistent improvements. For example, on CIFAR-10, IEC reduces FID from 4.70 to 3.96 when N = 3. Similar trends are observed for larger N, with FID reduced from 17.21 to 14.58 when N = 15. On LSUN-Churchs, IEC reduces FID from 25.27 to 22.42 at N = 15, and on LSUN-Bedrooms, from 26.09 to 16.91 at N = 10.

**Combined With Quantization-Caching.** The results of IEC on CacheQuant (Liu et al., 2025b), which integrates quantization and feature caching, are shown in Tab. 3 and Tab. 4. On CIFAR-10, IEC achieves consistent improvements. For instance, under W8A8 with N = 3, FID is reduced from 4.61 to 3.93. With N = 15, FID improves from 13.42 to 10.77. Similar improvements are observed under W4A8. For example, at N = 10, FID improves from 11.36 to 10.69. On LSUN-Churchs, for W8A8 and N = 3, IEC reduces FID from 3.66 to 3.39. At N = 15, FID drops from 9.47 to 6.90. Under W4A8, IEC improves FID by 0.23, 0.45, 1.06, and 1.78 for N = 3, 5, 10, and 15, respectively. On LSUN-Bedrooms, IEC also provides consistent gains. For example, under W8A8 with N = 2, FID improves from 8.85 to 7.51. As shown in Tab. 4, on ImageNet, IEC enhances performance across all baselines. For instance, under W8A8 with N = 10, FID improves from 4.68 to 4.15 and IS from 184.38 to 196.20. At N = 20, FID decreases from 7.21 to 6.53, and IS increases from 160.68 to 169.71. Under W4A8, IEC also improves performance. For example, at N = 10, FID improves from 6.90 to 6.50 and IS from 158.27 to 161.86, demonstrating IEC's robustness in low-precision scenarios. On MS-COCO, IEC also brings improvements. For example, under W8A8 with N = 10, FID improves from 23.65 to 23.36, IS from 36.71 to 37.02, and CLIP Score from 26.41 to 26.45. At N = 5, FID improves from 23.74 to 22.83, IS from 39.81 to 40.91, and CLIP Score from 26.87 to 26.94.

In summary, across all datasets and efficiency techniques, the integration of IEC consistently enhances performance, demonstrating its effectiveness.

Table 4: Results of combining IEC with CacheQuant (Liu et al., 2025b) (Denoted as CacheQ) on ImageNet and MS-COCO datasets.

| W/A | ImageNet 256 × 256 (T=250) | | |
| --- | --- | --- | --- |
| | Method | FID ↓ | IS ↑ |
| | LDM-4 | 3.37 | 204.56 |
| 8/8 | CacheQ-N=10 | 4.68 | 184.38 |
| | +IEC | **4.15** | **196.20** |
| | CacheQ-N=15 | 5.51 | 174.81 |
| | +IEC | **5.18** | **182.30** |
| | CacheQ-N=20 | 7.21 | 160.68 |
| | +IEC | **6.53** | **169.71** |
| 4/8 | CacheQ-N=10 | 6.90 | 158.27 |
| | +IEC | **6.50** | **161.86** |
| | CacheQ-N=15 | 9.40 | 139.64 |
| | +IEC | **8.66** | **144.41** |
| | CacheQ-N=20 | 12.65 | 124.13 |
| | +IEC | **11.10** | **130.01** |

| W/A | MS-COCO 256 × 256 (T=50) | | |
| --- | --- | --- | --- |
| | Method | FID ↓ | IS ↑ | CLIP Score ↑ |
| | PLMS | 22.41 | 41.02 | 26.89 |
| 8/8 | CacheQ-N=10 | 23.65 | 36.71 | 26.41 |
| | +IEC | **23.36** | **37.02** | **26.45** |
| | CacheQ-N=5 | 23.74 | 39.81 | 26.87 |
| | +IEC | **22.83** | **40.91** | **26.94** |
| 4/8 | CacheQ-N=10 | 26.63 | 34.57 | 26.23 |
| | +IEC | **24.82** | **36.18** | **26.25** |
| | CacheQ-N=5 | 23.85 | 39.53 | 26.78 |
| | +IEC | **23.80** | **40.27** | **26.80** |

## 4.4 OVERHEAD DISCUSSION

Table 5 presents the overhead and FID when combining our IEC with timestep-wise quantization (Liu et al., 2024b) and Deepcache (Ma et al., 2024b) on the CIFAR-10 dataset. Here, we measure the time overhead by comparing it with the baseline without IEC. For W8A8 quantization, IEC achieves substantial improvements in FID with minimal overhead. For instance, selectively applying IEC to $\pm 1/10$ or $\pm 1/20$ of the steps achieves improved FID of 3.88 and 3.97 with only 20% and 10% overhead, respectively. In the case of DeepCache, the IEC is only applied to the non-cached timesteps,

thereby its overhead is minimal. For example, applying IEC to all timesteps yields a FID of 7.77 with only 14% overhead, while applying IEC to $\pm 1/10$ or $\pm 1/20$ reduces the FID to 9.58 and 9.55, respectively, with overheads as low as 2.8% and 1.4%. These results demonstrate that IEC provides a flexible trade-off between efficiency and generation quality, making it easy to use and suitable for real-world applications.

Table 5: Overhead of combining IEC with timestep-wise quantization (Liu et al., 2024b) and Deepcache (N=10) (Ma et al., 2024b) on CIFAR-10. "Naive + T" indicates increasing "T" iteration count.

| CIFAR-10 32 $\times$ 32 (T=100) | | | | CIFAR-10 32 $\times$ 32 (T=100) | | |
|---|---|---|---|---|---|---|
| W/A | FID $\downarrow$ | Time Overhead | | Method | FID $\downarrow$ | Time Overhead |
| DDIM | 4.19 | - | | DDIM | 4.19 | - |
| W8A8 | 4.32 | 0% | | Deepcache-N=10 | 9.74 | 0% |
| Naive + 100 | 3.99 | 100% | | Naive + 14 | 8.58 | 14% |
| All Step (K=1) | 3.76 | 100% | | All Step (K=1) | 7.77 | 14% |
| $\pm 1/4$ | 3.85 | 50% | | $\pm 1/4$ | 7.92 | 7% |
| $\pm 1/8$ | 3.86 | 25% | | $\pm 1/8$ | 9.31 | 4.2% |
| $\pm 1/10$ | 3.88 | 20% | | $\pm 1/10$ | 9.58 | 2.8% |
| $\pm 1/20$ | 3.97 | 10% | | $\pm 1/20$ | 9.55 | 1.4% |

### 4.5 COMPARISON WITH NAIVELY ADDING ITERATION

In this subsection, we compare IEC with two alternative approaches: naively increasing the number of iterations in timestep-wise quantization and DeepCache. Specifically, simply adding 100% more iterations to the timestep-wise quantization method yields an FID score of 3.99, which is worse than our IEC with 100% overhead (3.76 FID) and even inferior to IEC with only 10% overhead (3.97 FID), as shown in Tab. 5. Similarly, increasing the iteration count by 14% for Deepcach-N=10 leads to an FID of 8.58, which is also worse than the 7.77 FID achieved by our IEC. These results demonstrate that merely increasing the number of iterations does not lead to satisfactory performance and justify the effectiveness of IEC.

## 5 DISCUSSION

The analysis in Sec. 3.1 suggests that more robust models can be achieved by modifying the scheduling schemes of $A_t$ and $B_t$, or by explicitly fine-tuning the model to control the norm of the Jacobian. Additionally, as discussed in Sec. 4.2, identifying a small subset of critical timesteps for applying the proposed IEC can significantly reduce inference overhead while still improving generation quality. Finally, this paper presents a conceptual and experimental validation of the test-time method for diffusion, while further exploring other diffusion models, samplers, and efficiency techniques remains an interesting topic. Although these directions are promising, we leave their exploration to future work due to current resource limitations.

## 6 CONCLUSION

In this paper, we address the challenge of improving the performance of efficient diffusion models at test-time. We begin by analyzing the error accumulation in such models and show that these errors can grow exponentially during the denoising process. To mitigate this, we introduce Iterative Error Correction (IEC), a simple yet effective method that iteratively refines the model's output to suppress error propagation. Our theoretical analysis demonstrates that IEC converges to a fixed-point solution, reducing the rate of error growth from exponential to linear. As a plug-and-play, model-agnostic method, IEC offers a flexible trade-off between performance and efficiency, allowing users to adapt it to various practical scenarios. We validate IEC through extensive experiments, showing consistent performance gains across various datasets, efficiency techniques, and diffusion model architectures.

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

# A    APPENDIX

## A.1    ALGORITHM PROCESS

Alg. 1 presents the overall process of the proposed IEC.

---

**Algorithm 1** Iterative Error Correction (IEC)

---

1: **Input:** $x_t$, timestep $t$, hyperparameter $\lambda$, threshold $\tau$, max iterations $K$
2: **Output:** Final corrected estimate $x^*_{t-1}$
3: Initialize $x^{(0)}_{t-1}$ using Eq. 6
4: **while** $k < K$ **do**
5:     Obtain $x^{(k+1)}_{t-1}$ using Eq. 10
6:     **if** $\|x^{(k+1)}_{t-1} - x^{(k)}_{t-1}\| < \tau$ **then**
7:         **break**
8:     **end if**
9:     $k = k + 1$
10: **end while**
11: **Return:** $x^*_{t-1} = x^{(k+1)}_{t-1}$

---

## A.2    EXPLANATION OF THE MISS OF W4A8 RESULTS ON LSUN-BEDROOMS

Applying W4A8 quantization on the LSUN-Bedrooms dataset results in severe performance degradation. For instance, with CacheQ (N=10) under W4A8 settings, the baseline FID on LSUN-Bedrooms degrades to 156.3. Due to this extreme baseline failure, we omit these specific configurations from the main comparative results (Tab. 1 and Tab. 3). However, this severe degradation provides a valuable testbed for evaluating the recovery capabilities of our IEC under extreme conditions. Notably, even with such a damaged baseline, applying IEC drastically improves the FID from 156.3 to 52.6, demonstrating its capacity to recover signal from heavily quantized models.

## A.3    DISCUSSION OF IEC'S BOUNDARIES

IEC is designed to correct the denoising trajectory deviations resulting from the error in efficiency techniques. However, in cases of total mode collapse or an under-trained model, where the model's prediction becomes entirely random, since IEC cannot hallucinate correct features that the model has totally lost.

## A.4    APPLYING IEC ACROSS VARYING SETTINGS

We further explore the IEC's effectiveness across varying DDIM sampling steps and hyperparameters in efficiency methods in Fig. 4. It can be seen that for any given setting, IEC consistently presents better performance, demonstrating its effectiveness.

## A.5    VISUALIZATION

In this section, we provide visualization comparisons between the proposed IEC and the corresponding baseline. Fig. 5 and Fig. 6 show the visualization results of Stable Diffusion on the COCO dataset. Fig. 7, Fig. 8, and Fig. 9 show the visualization results of LDM-8 on the LSUN-Churchs datasets. Fig. 10, Fig. 11, and Fig. 12 show the visualization results of LDM-4 on the LSUN-Bedrooms datasets.

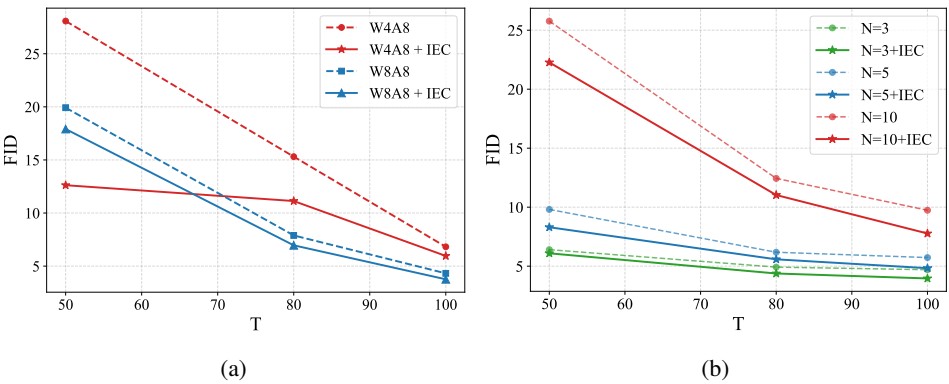

(a)                      (b)

Figure 4: (a) Performance comparison of quantization (W4A8/W8A8) with and without IEC on the CIFAR-10 dataset across varying sampling steps. (b) Performance comparison of DeepCache ($N \in \{3, 5, 10\}$) combined with IEC across varying sampling steps.

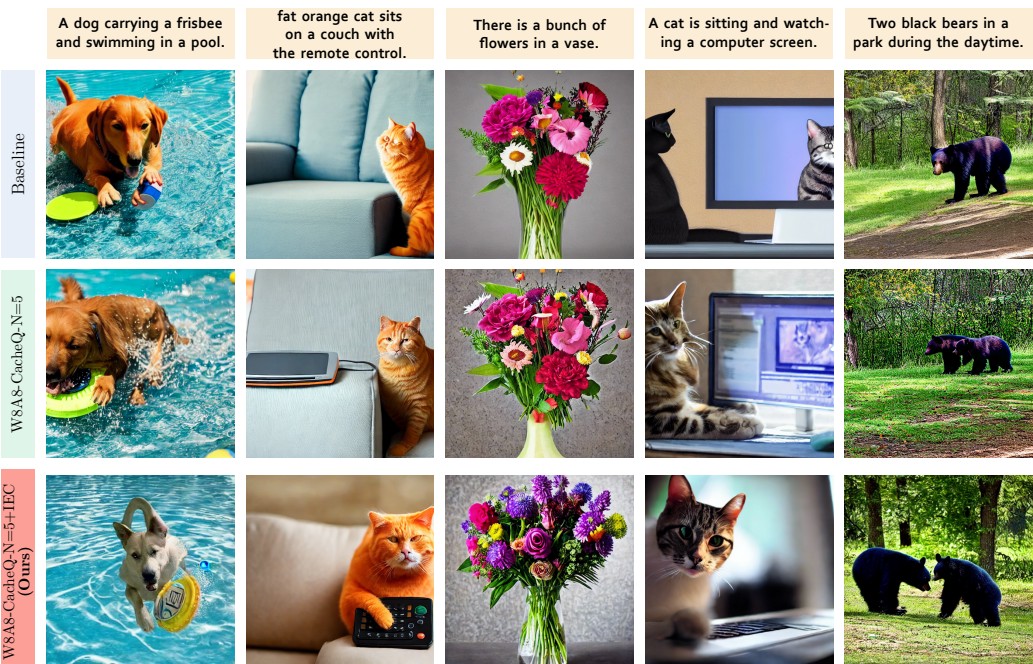

Figure 5: Qualitative comparison of Stable Diffusion on the COCO dataset: Baseline *vs.* CacheQuant (W8A8, N=5) with and without IEC.

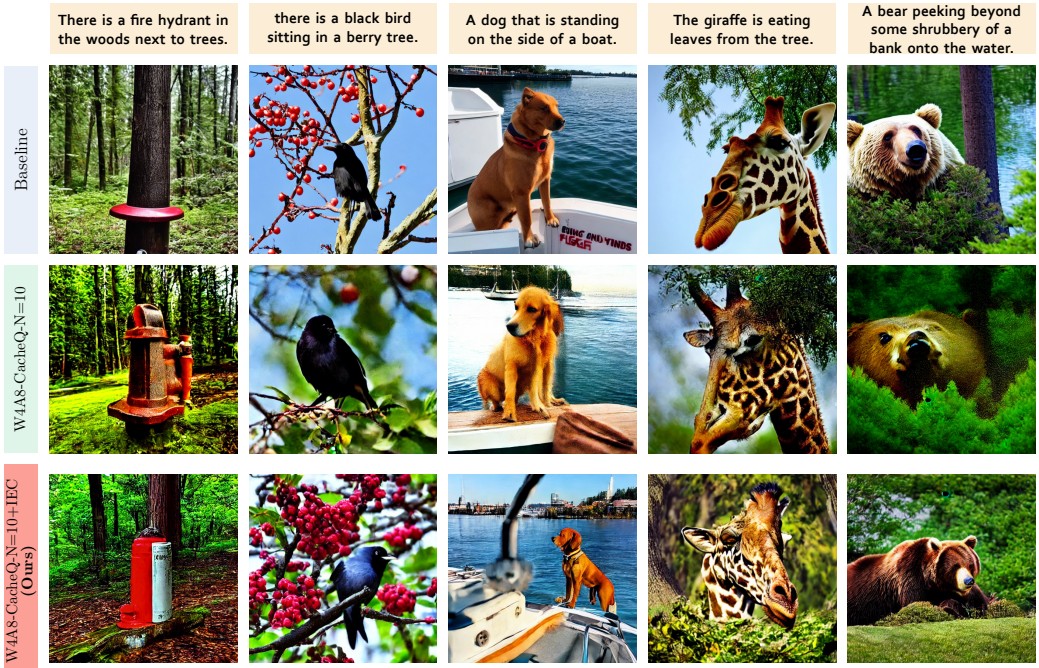

Figure 6: Qualitative comparison of Stable Diffusion on the COCO dataset: Baseline *vs.* CacheQuant (W4A8, N=10) with and without IEC.

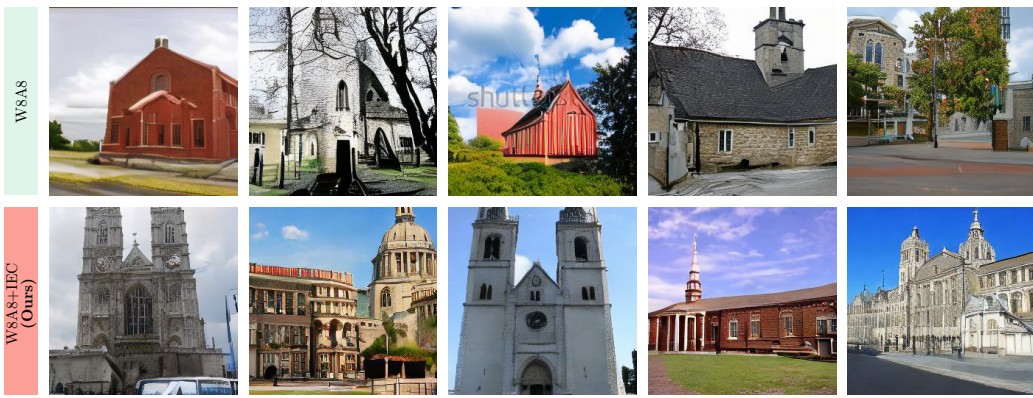

Figure 7: Qualitative comparison of LDM-8 on the LSUN-Churchs dataset: W8A8 *vs.* W8A8+IEC.

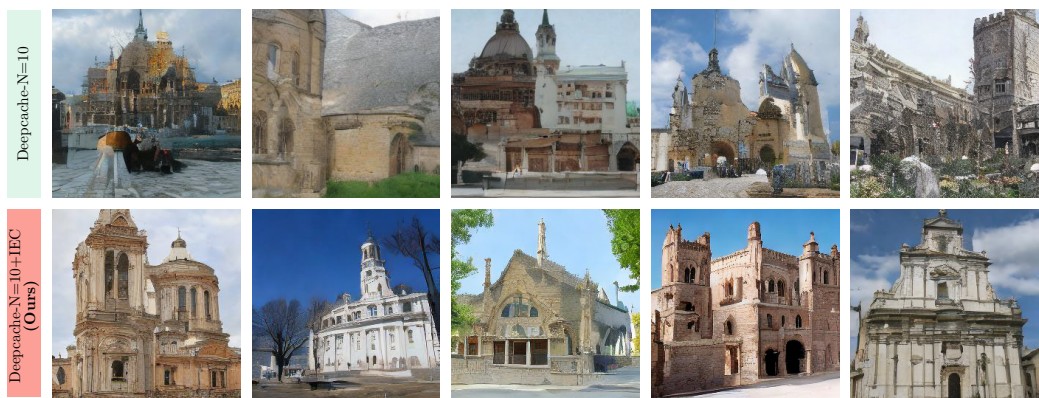

Figure 8: Qualitative comparison of LDM-8 on the LSUN-Churchs dataset: DeepCache (N=10) *vs.* DeepCache+IEC (N=10) .

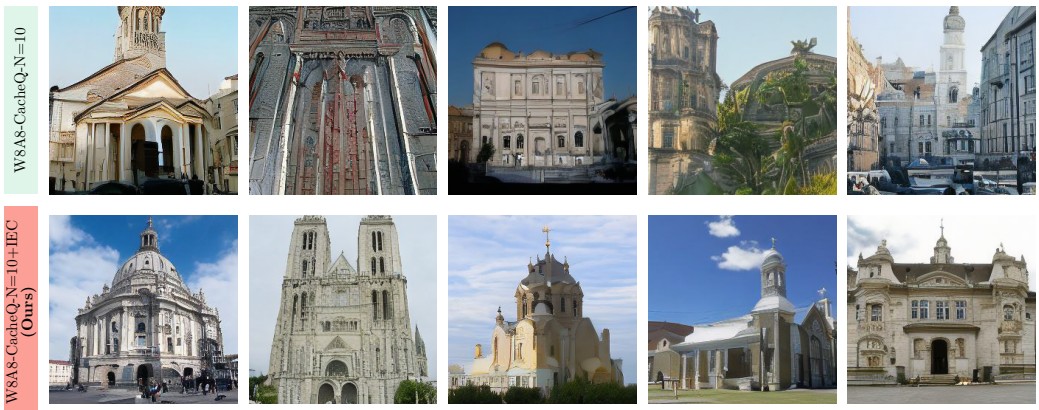

Figure 9: Qualitative comparison of LDM-8 on the LSUN-Churchs dataset: CacheQuant (W8A8, N=10) *vs.* CacheQuant+IEC (W8A8, N=10) .

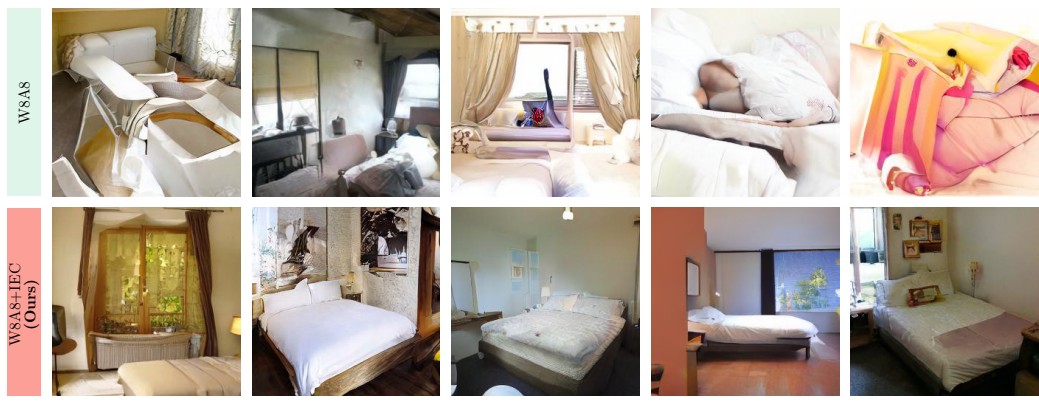

Figure 10: Qualitative comparison of LDM-4 on the LSUN-Bedrooms dataset: W8A8 *vs.* W8A8+IEC.

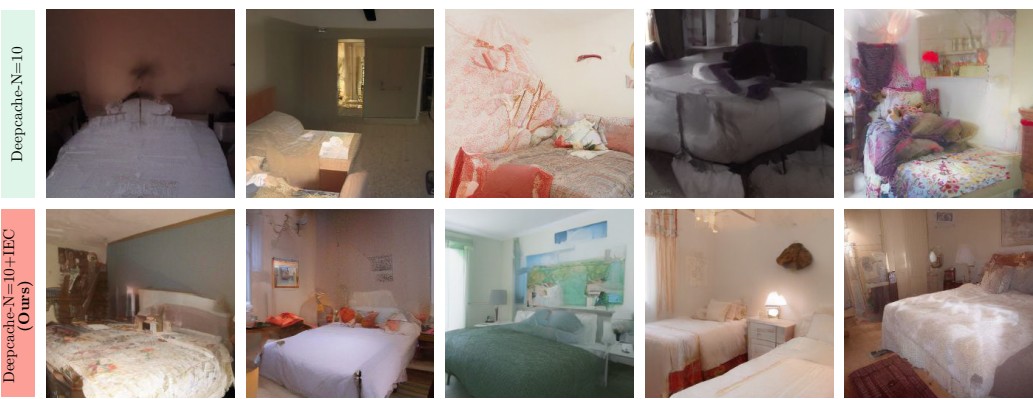

Figure 11: Qualitative comparison of LDM-4 on the LSUN-Bedrooms dataset: DeepCache (N=10) *vs.* DeepCache+IEC (N=10) .

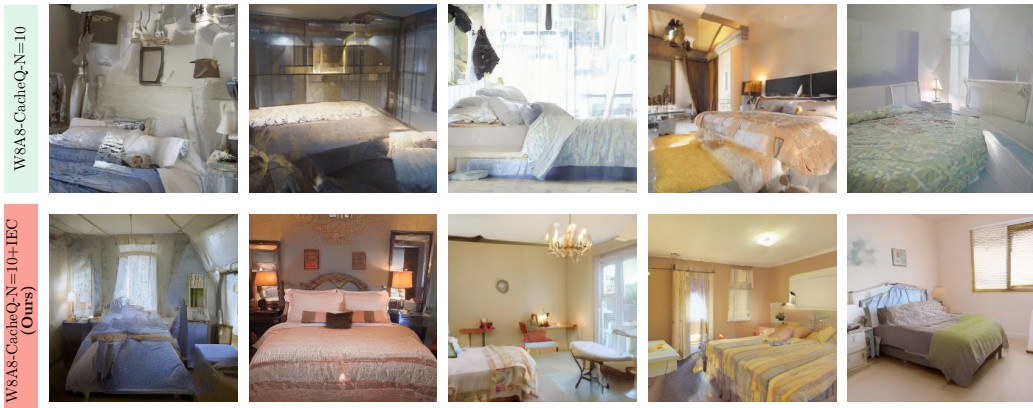

Figure 12: Qualitative comparison of LDM-4 on the LSUN-Bedrooms dataset: CacheQuant (W8A8, N=10) *vs.* CacheQuant+IEC (W8A8, N=10) .

