# OpenReview forum: "Test-Time Iterative Error Correction for Efficient Diffusion Models"
_ICLR.cc/2026/Conference — ICLR 2026 Poster_

### Official Review · Reviewer_roVY · 2025-10-26

**Soundness:** 3
**Presentation:** 3
**Contribution:** 3
**Rating:** 6
**Confidence:** 3

**Summary:**

This paper proposes a novel inference-time method called Iterative Error Correction (IEC) to enhance the performance of efficient diffusion models that often degrade in quality due to approximations introduced by quantization, feature caching, or other efficiency techniques. The authors introduce IEC, a test-time, plug-and-play refinement procedure that corrects inference-time errors without modifying model parameters. IEC is evaluated on multiple diffusion models (DDPM, LDM, and Stable Diffusion) and efficiency schemes (quantization, feature caching, and hybrid quantization-caching), across datasets such as CIFAR-10, LSUN-Church, LSUN-Bedroom, ImageNet, and MS-COCO.

**Strengths:**

1. The paper introduces Iterative Error Correction (IEC), a new test-time method for improving diffusion model performance without retraining or architecture changes. This is highly practical because deployed diffusion models—especially quantized or cached versions—are often immutable, and IEC offers a plug-and-play solution that directly operates at inference time.

2. The authors provide a clear mathematical analysis of error accumulation in efficient diffusion models and show that approximation errors grow exponentially across timesteps. They then rigorously prove that IEC reduces this exponential growth to linear error propagation through a fixed-point convergence analysis. This theoretical grounding strengthens the credibility and generality of the proposed method.

3. Since IEC operates purely at test time, it does not require access to the original full-precision model or re-application of the efficiency pipeline. This feature is particularly valuable for real-world scenarios where deployed models are frozen or stored under strict hardware and policy constraints.

**Weaknesses:**

1. The experiments are limited in CNN-based methods. The effectiveness of the proposed methods are not validated in Transformer-based methods such as DiT and Flux.

2. The theoretical and experimental analysis builds on top of DDIM sampler. I wonder if this error reduction strategy is compatible with other samplers such as DPM-Solver?

3. Some critical implementation details are missing such as the exact version of SD.

**Questions:**

1. Would this strategy also applicable in other acceleration methods such as adaptive computation[1]? Could you provide some empirical or theoretical analysis for this?

[1] Tang S, Wang Y, Ding C, et al. Adadiff: Accelerating diffusion models through step-wise adaptive computation[C]//European Conference on Computer Vision. Cham: Springer Nature Switzerland, 2024: 73-90.

---

> ### Author Response · Authors · 2025-11-20
> **Response to Reviewer roVY**
>
> Thank you very much for your valuable comments!
>
> ---
>
> *W1. The experiments are limited in CNN-based methods. The effectiveness of the proposed methods are not validated in Transformer-based methods such as DiT and Flux.*
>
> **Response to W1:**
>
> IEC is fundamentally model-agnostic and naturally extends to transformer-based architectures. To demonstrate this, we conducted additional experiments on DiT-XL/2 (ImageNet $256\times256$) under both W8A8 and W4A8 quantization settings, combined with Deepcache (N=2 for T=20; N=5 for T=50). The Table below shows that IEC significantly recovers performance. Notably, for DiT-XL/2 (W4A8, T=50, N=5), IEC reduces FID from 9.42 to 5.53, a 41% improvement, effectively rescuing the model from severe degradation.
>
> | Model | W/A | N| FID (smaller is better) | IS (higher is better)|
> | -- | -- | -- |-- |-- |
> | DiT-XL/2 (T=50) | 8/8 | 5 | 6.71 | 193.02|
> | DiT-XL/2 (T=50) + IEC (ours)| 8/8 | 5 | **4.56** | **222.7**|
> |-|
> | DiT-XL/2 (T=50) | 4/8 | 5 | 9.42 | 157.2|
> | DiT-XL/2 (T=50) + IEC (ours)| 4/8 | 5 | **5.53** |**192.8**|
> |-|
> | DiT-XL/2 (T=20) | 8/8 | 2 |7.63|184.2|
> | DiT-XL/2 (T=20) + IEC (ours)| 8/8 | 2 |**6.20**|**198.4**|
> |-|
> | DiT-XL/2 (T=20) | 4/8 | 2 | 10.3|158.4|
> | DiT-XL/2 (T=20) + IEC (ours) | 4/8 | 2 | **9.01**|**167.4**|
>
>
> ---
>
> *W2. The theoretical and experimental analysis builds on top of DDIM sampler. I wonder if this error reduction strategy is compatible with other samplers such as DPM-Solver? *
>
>
> **Response to W2:**
>
>  We thank the reviewer for this insightful question. The first‑order DPM‑Solver is essentially equivalent to the DDIM sampler used in our analysis. Since our theoretical framework is proven under DDIM, it naturally applies to DPM‑Solver‑1 as well. For higher‑order DPM‑Solver variants, IEC functions as an orthogonal corrector. High-order solvers optimize the trajectory path (reducing discretization error) but also suffer error if the model is quantized/cached. We consider IEC is complementary to these solvers and view this as a promising direction for future work.
>
> ---
>
> *W3. Some critical implementation details are missing such as the exact version of SD.*
>
>
> **Response to W3:**
>
> Thank you very much! For Stable Diffusion experiments, we utilized the CompVis codebase and the official v1.4 checkpoint. We have added this information to our paper (Sec 4.1).
>
> ---
>
> *Q1. Would this strategy also applicable in other acceleration methods such as adaptive computation[1]? Could you provide some empirical or theoretical analysis for this?*
>
> **Response to Q1:**
>
> We thank the reviewer for pointing out this relevant work. We confirm that IEC is theoretically applicable to adaptive computation methods like AdaDiff. Our theoretical framework in Sec 3.1 models the efficient model's prediction as a perturbed version of the ideal prediction: $\tilde\epsilon_\theta(x_t, t) = \epsilon_\theta(x_t, t) + \epsilon^\delta$ where $\epsilon^\delta$ represents the approximation error. In AdaDiff [1], the method accelerates inference by dynamically skipping residual blocks or layers based on timestep difficulty. This introduces a structural approximation error, which can also be formulated as $\epsilon^\delta$. Since our derivation of the error propagation (Eq. 8) and the correction rule (Eq. 10) depends only on the existence of the deviation $\epsilon^\delta$, IEC can directly treat the layer-skipping error as a target for correction. Moreover, we have cited this paper in our revised paper for a comprehensive related work.

---

### Official Review · Reviewer_cbcn · 2025-10-30

**Soundness:** 2
**Presentation:** 3
**Contribution:** 2
**Rating:** 4
**Confidence:** 3

**Summary:**

This paper proposes Iterative Error Correction (IEC), a test-time method to improve the quality of efficient diffusion models suffering from approximation errors (e.g., from quantization or feature caching). The authors analyze error propagation in DDIM sampling and claim errors accumulate exponentially. IEC iteratively refines outputs at each timestep using a fixed-point iteration scheme, theoretically proven to reduce error growth from exponential to linear. Experiments on CIFAR-10, LSUN, ImageNet, and MS-COCO demonstrate FID improvements when combining IEC with quantization and caching methods.

**Strengths:**

1. Clear problem motivation (deployed models can't be easily modified) thus a training free method is needed.
2. The ability to apply IEC to selected timesteps (±1/10, ±1/20) provides efficiency options, though this flexibility is expected from any refinement method
3. IEC shows gains across all tested scenarios, demonstrating broad applicability
4. The paper is generally easy to follow with good visual aids

**Weaknesses:**

Fundamentally Unfair Comparison:
1. Baseline uses 100 forward passes, while IEC with K=1 applied to all T=100 steps uses 200 forward pass.
2. Missing crucial baseline: what is the FID with T=200 steps without IEC?
3. This is like claiming "Method A is better than Method B" when Method A uses 2× computation

IEC is Multi-Step Refinement essentially
1. Eq. 10 is mathematically equivalent to: repeatedly calling the denoising function with corrected inputs
2. This is essentially running multiple sampling steps per timestep
3. Authors admit in A.2 that "naively adding 100% iterations" achieves FID 3.99, close to IEC's 3.76

**Questions:**

1. What is the FID when using T=200 sampling steps without IEC, matching the computational budget of T=100 with IEC (K=1)?
2. How does IEC compare to other methods that increase sampling quality with more compute (e.g., DPM-Solver++ with more steps)?
3. Can you provide rigorous proof that errors grow exponentially without IEC? The current analysis only shows they propagate through products of matrices.
4. How does IEC compare to consistency models or other iterative refinement approaches?

---

> ### Author Response · Authors · 2025-11-20
> **Response to Reviewer cbcn-part1/2**
>
> We highly appreciate your comments! In the following, we addressed the raised concerns point-by-point.
>
> ---
>
> *W1 The fairness of comparison.*
>
> *W1.1 Baseline uses 100 forward passes, while IEC with K=1 applied to all T=100 steps uses 200 forward pass.*
>
> *W1.2 Missing crucial baseline: what is the FID with T=200 steps without IEC?*
>
> *W1.3 This is like claiming "Method A is better than Method B" when Method A uses 2× computation*
>
> **Response to W1.1 & W1.3:**
>
> Thank you very much! We respectfully submit that comparing IEC to simply doubling sampling steps ($T=200$) is not an equivalent comparison in the context of efficient deployment, as it overlooks critical deployment constraints and retraining costs.
> 1. For efficient models (e.g., quantization or caching), changing the schedule to $T=200$ is not free. It often requires expensive **re-calibration or re-training** to adapt quantization/caching parameters to the new timestep trajectory. In contrast, IEC is a training-free, plug-and-play solution designed for frozen, deployed models, providing a remedy where retraining is technically or computationally infeasible.
> 2. Even solely considering inference compute, IEC demonstrates superior efficiency. As shown in Sec. A.1 (in the revised paper, the Sec A.1 has been moved to Sec 4.4), applying IEC to only 10% of steps incurs a total cost of 1.1x yet achieves an FID of 3.97, outperforming the naive $T=200$ baseline (Cost 2.0x, FID 3.99). Crucially, increasing $T$ does not change the inherent error accumulation trend, whereas IEC explicitly corrects it. This proves IEC utilizes the computational budget more effectively than naive scaling.
> 3. Furthermore, IEC acts as a modular plugin with dynamically adjustable cost ($K$ iterations, percentage of applied steps). This allows users to tune the compute-quality trade-off at runtime, which is a capability that fixed-schedule deployed models fundamentally lack.
>
> **Response to W1.2:**
>
> Thank you. Following your advice, we conducted experiments using $T=200$ for W8A8 LSUN-Churchs and LSUN-Bedrooms. The $T=200$ baseline achieves an FID of 4.74 on LSUN-Churchs and 8.56 on LSUN-Bedrooms. Both are still inferior to applying IEC on $T=100$ (3.29 for LSUN-Churchs and 7.78 for LSUN-Bedrooms, as shown in Table 1). We consider that this is because simply adding a step also accumulates more quantization error across time, which ultimately hampers the generation quality. These results further confirm the superiority of IEC even against a double-compute baseline.
>
> ---
>
> *W2 Multi-Step Refinement*
>
> *W2.1 Eq. 10 is mathematically equivalent to: repeatedly calling the denoising function with corrected inputs.*
>
> *W2.2 This is essentially running multiple sampling steps per timestep*
>
> *W2.3 Authors admit in A.2 that "naively adding 100% iterations" achieves FID 3.99, close to IEC's 3.76*
>
> **Response to W2.1 & W2.2:**
>
> We agree with the reviewer’s observation that IEC mechanically involves repeatedly calling the denoising function. However, we respectfully argue that this simplicity conceals a deliberate theoretical construction. The specific formulation of Eq. 10 is not an arbitrary repetition, but a strategic design derived to guarantee
> The core insight lies in recognizing that complex, exponential error accumulation can be resolved through a mathematically elegant Fixed-Point Iteration (Eq. 10). By reusing the pre-trained denoising function as a contraction mapping operator, we prove that a simple iterative call is sufficient to reduce error growth from exponential to linear. Crucially, without this specific mathematical design, simple repeated sampling lacks convergence guarantees and cannot achieve error suppression.
>
> **Response to W2.3:**
>
> The reviewer notes that naive doubling (2.0x cost) achieves FID 3.99, close to IEC's 3.76. However, this comparison misses two critical points. First, as shown in Sec A.1 (in the revised paper, Sec A.1 has been moved to Sec 4.4), IEC achieves the quality of the $2.0\times$ baseline while requiring 90% less additional compute (0.1x vs 1.0x extra cost). This proves that correcting specific errors is far more efficient than simply increasing sampling steps. Second, the naive baseline ($T=200$) represents a static, immutable cost. Once an efficient model is deployed, its inference schedule is typically frozen to ensure calibration stability. Users cannot dynamically "pay more" for better quality. In contrast, IEC allows for a flexible test-time trade-off, enabling runtime elasticity based on latency constraints—a fundamental capability that fixed-step baselines lack.

---

> ### Author Response · Authors · 2025-11-20
> **Response to Reviewer cbcn-part2/2**
>
> *Q1. What is the FID when using T=200 sampling steps without IEC, matching the computational budget of T=100 with IEC (K=1)?*
>
> **Response to Q1:** Please see the response to W1.2.
>
> ---
>
> *Q2. How does IEC compare to other methods that increase sampling quality with more compute (e.g., DPM-Solver++ with more steps)?*
>
> **Response to Q2:**
>
> Thank you for the question. Methods like DPM‑Solver++ improve sampling accuracy by using more or higher‑order steps, assuming the underlying diffusion model is accurate. In efficient diffusion models, however, the dominant degradation comes from model approximation errors (quantization/caching), which cannot be corrected by increasing solver steps (see response to W2.3).
> In contrast, IEC directly suppresses test‑time approximation errors via a fixed‑point refinement mechanism. Thus, in principle, IEC is complementary to DPM‑Solver++ if desired, but our focus is on enhancing efficient models whose quality cannot be recovered simply by using more sampling steps.
>
> ---
>
> *Q3. Can you provide rigorous proof that errors grow exponentially without IEC? The current analysis only shows they propagate through products of matrices.*
>
> **Response to Q3:**
>
> Thank you! Our claim of exponential growth is supported by theoretical derivation and empirical verification. First, our derivation in Eq. 9 shows that the final error is a sum of terms scaled by the norm of matrices, whose norm is consistently $> 1$ throughout the diffusion process as shown in Figure 1(a). Furthermore, Figure 2 and the new table below provide practical accumulated error over time. The rapid, non-linear rise in error magnitude empirically confirms the exponential growth predicted by our theory.
>
>  | LSUN-Churchs |T=1|T=20| T=40| T=60| T=80| T=100|
> |--|--|--|--|--|--|--|
> |W8A8 |0.007|0.205|0.592|2.72|30.22|141.5|
> |Deepcache (N=10) |5.89e-06|0.15|2.51| 14.69|64.89|188.6|
> |-|
> |LSUN-Bedrooms|T=1 | T=20| T=40| T=60| T=80| T=100|
> |W8A8 |0.04|1.35|7.12|38.06|143.8|428.1|
> |Deepcache (N=10) |1.65e-05|0.10|2.88|40.00|204.7|598.7|
>
> ---
>
> *Q4. How does IEC compare to consistency models or other iterative refinement approaches?*
>
>
> **Response to Q4:**
>
> We distinguish IEC from Consistency Models (CM) based on training requirements. Consistency models and related refinement methods typically require training or distillation to construct new fast generators. In contrast, IEC instead is a pure test‑time method that requires no retraining, no architecture changes, and is designed for already‑deployed efficient models to suppress errors in efficient diffusion models, where the original weights may not be available. Thus, their mechanisms are fundamentally different and address different stages of the model lifecycle.

---

> > ### Comment · Reviewer_cbcn · 2025-11-26
> >
> > Thank you for providing additional results and explanations. Most of my concerns are addressed in the rebuttal. I have updated my rating.

---

> > > ### Author Response · Authors · 2025-11-27
> > >
> > > Thank you very much! We highly appreciate the effort made by the reviewer and are glad to improve our paper according to your valuable suggestions.

---

### Official Review · Reviewer_NVsH · 2025-10-31

**Soundness:** 3
**Presentation:** 2
**Contribution:** 3
**Rating:** 6
**Confidence:** 4

**Summary:**

This paper addresses the performance degradation problem in efficient diffusion models, which arises from approximation errors introduced by techniques like quantization and feature caching. It shows that these errors can accumulate exponentially throughout the sampling.

The core contribution is a test-time-scaling-like method called Iterative Error Correction (IEC). It requires no retraining or architectural changes, and operates by introducing an inner-loop fixed-point iteration at each sampling timestep to refine the model's output, which reduces the error accumulation from exponential to linear.

The paper provides both theoretical and empirical analysis showing that IEC can reduce the error accumulation. Experiments on various models (DDPM, LDM, SD), datasets (CIFAR, LSUN, ImageNet), and efficiency techniques (quant, cache, quant+cache) demonstrate that IEC consistently improves generation quality.

**Strengths:**

-- Problem: It targets the practical, post-deployment scenario for efficient models, which is somewhat under-explored.

-- Idea: It is related to test-time scaling, and potentially provides another new dimension in which we can scale (the inner-loop iteration at each timestep).

-- Analysis: The theoretical analysis is of good quality. It is also appreciated that the authors try to validate their method on multiple models and settings.

**Weaknesses:**

-- The theoretical framework seems general to not be limited to errors from efficiency techniques.
It is unclear how the definition of $\tilde x_t = x_t + \delta_t$ and $\tilde\epsilon_\theta = \epsilon_\theta + \epsilon_\theta^\delta$ are necessarily linked to quantization or caching.
It seems potentially applicable to any diffusion model, where $x_t$ and $\epsilon_\theta$ correspond to an ideal denoiser (please refer to Figure 1 in EDM).
It would strengthen the paper if the authors could show the method can also improve regular diffusion models (e.g., trained with fewer iterations and thus have imperfections).

--The compute-performance trade-off is not fully explored.
According to my understanding, the total inference compute is determined by at least three factors: (1) DDIM sampling steps (2) hyper-parameters in efficiency methods (3) hyper-parameters in IEC. Each of these can be adjusted to trade more computation for better performance.
While IEC shows improvements when (1) and (2) are fixed, it is unclear whether it provides a better Pareto frontier in the broader context. For example, a real, comprehensive compute-performance plot (please refer to Figure 9 in EDM2 and Figure 10 in DiT) with respect to all these factors would be helpful. The current discussion in A.1 and A.2 is insightful but seems too simple.

--Theoretical and empirical analysis is based only on DDPM/DDIM variants, without showing applicability to modern flow-based models.

--Presentation
Tables: Table 1 and Table 3 take too much space, and their subtables could be merged like Table 2. Additionally, it is unclear why results for certain settings are omitted (e.g., W4A8 in Table 1 for LSUN-Bedrooms, and W8A8 in Table 3). I assume W8A8 and W4A8 are both meaningful in both tables.
Section A.2: some cited data (e.g., 3.99, 8.58) do not appear in Table 5. The overhead and trade-off (claimed in Abstract) analysis is important and should be moved to the main text.
Typo: "ImageNet 256x25" in Table 4.
Citation: most should be in parenthesis using \citep{}.

EDM: Elucidating the Design Space of Diffusion-Based Generative Models, NeurIPS 2022

EDM2: Analyzing and Improving the Training Dynamics of Diffusion Models, CVPR 2024

DiT: Scalable Diffusion Models with Transformers, ICCV 2023

**Questions:**

Please refer to the weaknesses about generalization and compute-performance trade-off.

---

> ### Author Response · Authors · 2025-11-20
> **Response to Reviewer NVsH-part1/2**
>
> We thank the reviewer for these valuable suggestions!
>
> ---
>
> *W1. The theoretical framework seems general to not be limited to errors from efficiency techniques. It is unclear how the definition of $\tilde x_t = x_t + \delta_t$ and $\tilde\epsilon_\theta = \epsilon_\theta + \epsilon_\theta^\delta$  are necessarily linked to quantization or caching. It seems potentially applicable to any diffusion model, where $x_t $ and $\epsilon_\theta$ and correspond to an ideal denoiser (please refer to Figure 1 in EDM). It would strengthen the paper if the authors could show the method can also improve regular diffusion models (e.g., trained with fewer iterations and thus have imperfections).*
>
> **Response to W1:**
>
> IEC is designed as a trajectory corrector to mitigate inference-time noise (e.g., quantization or caching error $\epsilon^\delta$), ensuring the trajectory adheres to the learned prediction trajectory. That says, IEC corrects the $\tilde x_t/\tilde\epsilon_\theta $ to ideal $x_t / \epsilon_\theta$ by solving $\delta_t/\epsilon_\theta^\delta$.
> However, in under-trained models, the error is epistemic (a fundamental lack of knowledge). The trajectory itself is ill-defined. That says, the $x_t / \epsilon_\theta$ itself are ill-defined. Thus, in this situation, IEC cannot produce correct features that the model has not yet learned.
>
> To demonstrate this, following your advice, we conducted an experiment applying IEC to a diffusion model checkpoint trained for only 50k steps (under-trained). We observed that IEC provided minimal improvement in this specific setting. This confirms that IEC is specifically designed for efficiency diffusion model scenarios, where the base model is capable (converged) but impaired by efficiency techniques, rather than for correcting training deficiencies.
>
> ---
>
> *W2. The compute-performance trade-off is not fully explored. According to my understanding, the total inference compute is determined by at least three factors: (1) DDIM sampling steps (2) hyper-parameters in efficiency methods (3) hyper-parameters in IEC. Each of these can be adjusted to trade more computation for better performance. While IEC shows improvements when (1) and (2) are fixed, it is unclear whether it provides a better Pareto frontier in the broader context. For example, a real, comprehensive compute-performance plot (please refer to Figure 9 in EDM2 and Figure 10 in DiT) with respect to all these factors would be helpful. The current discussion in A.1 and A.2 is insightful but seems too simple.*
>
> **Response to W2:**
>
> We thank the reviewer for this critical suggestion. We have added Pareto frontier plots (Figure 4 in Sec A.4) comparing IEC against standard baselines across varying DDIM steps ($T$), quantization bit-widths, and caching intervals. The figure’s data are shown in the tables below. IEC consistently yields superior generation quality for any given setting. Moreover, we have cited this paper in our revised paper for a comprehensive related work.
>
> | Method | Quantization | T=50 | T=80 | T=100 |
> | :--- | :---: | :---: | :---: | :---: |
> | Baseline | W4A8 | 28.08 | 15.31 | 6.82 |
> | **+ IEC (Ours)** | W4A8 | **12.62** | **11.13** | **5.96** |
> | Baseline | W8A8 | 19.92 | 7.89 | 4.32 |
> | **+ IEC (Ours)** | W8A8 | **17.91** | **6.96** | **3.76** |
>
> | Cache Interval ($N$) | Method | T=50 | T=80 | T=100 |
> | :---: | :--- | :---: | :---: | :---: |
> | **$N=3$** | Baseline | 6.40 | 4.92 | 4.70 |
> | | **+ IEC** | **6.09** | **4.38** | **3.96** |
> | **$N=5$** | Baseline | 9.81 | 6.18 | 5.73 |
> | | **+ IEC** | **8.30** | **5.58** | **4.83** |
> | **$N=10$** | Baseline | 25.77 | 12.44 | 9.74 |
> | | **+ IEC** | **22.27** | **11.03** | **7.77** |
>
>
>
> ---
>
> *W3. Theoretical and empirical analysis is based only on DDPM/DDIM variants, without showing applicability to modern flow-based models.*
>
> **Response to W3:**
>
> We thank the reviewer for this crucial suggestion. We clarify that our method's applicability extends directly to flow-based models since the Euler update used in flow matching ($x_{t-1} = x_t + \Delta t \cdot v_\theta$) is a special case of the linear update rule ($x_{t-1} = A_t x_t + B_t \epsilon_\theta$) derived in our paper.
>
> To demonstrate IEC’s effectiveness, we conducted new experiments on OT-CFM on CIFAR-10, combining W8A8 quantization with DeepCache. The table below shows consistent improvements. For instance, on OT-CFM (T=50, N=5), IEC reduces FID from 31.13 to 19.41, a 37% improvement. This confirms IEC's effectiveness.
>
> | Model | W/A | N| FID (smaller is better) |
> | -- | -- | -- |-- |
> | OT-CFM (T=100) | 8/8 | 3 | 10.59 |
> | OT-CFM (T=100) + IEC (ours) | 8/8 | 3 | **9.23** |
> | OT-CFM (T=100) | 8/8 | 5 | 15.08 |
> | OT-CFM (T=100) + IEC (ours) | 8/8 | 5 | **11.69** |
> | OT-CFM (T=50) | 8/8 | 3 | 10.53 |
> | OT-CFM (T=50) + IEC (ours) | 8/8 | 3 | **9.71** |
> | OT-CFM (T=50) | 8/8 | 5 |  31.13 |
> | OT-CFM (T=50) + IEC (ours) | 8/8 | 5 | **19.41** |

---

> ### Author Response · Authors · 2025-11-20
> **Response to Reviewer NVsH-part2/2**
>
> *W4. Presentation Tables: Table 1 and Table 3 take too much space, and their subtables could be merged like Table 2. Additionally, it is unclear why results for certain settings are omitted (e.g., W4A8 in Table 1 for LSUN-Bedrooms, and W8A8 in Table 3). I assume W8A8 and W4A8 are both meaningful in both tables. Section A.2: some cited data (e.g., 3.99, 8.58) do not appear in Table 5. The overhead and trade-off (claimed in Abstract) analysis is important and should be moved to the main text. Typo: "ImageNet 256x25" in Table 4. Citation: most should be in parenthesis using \citep{}.*
>
> **Response to W4:**
>
> Thank you for the detailed feedback on the presentation!
>
> 1. Merge Tables: We have merged Table 1 and Table 3 as suggested to improve readability.
>
> 2. Different bit-width of LSUN-Bedrooms in Table 1/3: We apologize for the confusion regarding the typo. The ‘W4A8’ of LSUN-Bedrooms in Table 3 should be corrected to ‘W8A8’. The omission of W4A8 results for LSUN-Bedrooms was intentional because W4A8 makes a significant performance loss. For example, for CacheQ-N=10 with W4A8 on LSUN-Bedrooms, the baseline FID was 156.3, indicating severe degradation. Note that we strictly follow the original code to reproduce this result. However, we find that applying IEC drastically improved the FID to 52.6, demonstrating that IEC can significantly recover the signal even from a heavily damaged model. We have included this discussion in the appendix of the revised paper (Sec A.2).
>
> 3. Cited data in Section A.2: Thank you very much! We have added this data to Table 5 for clarity.
>
> 4. Overhead and trade-off: Thank you! We have added these results to the main paper. Specifically, in the revised paper, Sec A.1 and Sec A.2 have been moved to Sec 4.4 and Sec 4.5, respectively.
>
> 5. Typo and Citation: Thank you! We have corrected these in our revised paper.

---

> > ### Comment · Reviewer_NVsH · 2025-11-27
> >
> > I thank the authors for their detailed responses and the additional experiments. The authors have successfully addressed most of my concerns, and I am inclined to recommend acceptance.
> >
> > Additional suggestion:
> > Regarding the Pareto Frontier, the current presentation (using timesteps as the x-axis) could be improved. I suggest using the actual compute cost as the x-axis in the final version. This would comprehensively integrate the base inference cost, the reduction from efficiency techniques, and the introduced IEC overhead, offering a fair and rigorous comparison.

---

> > > ### Author Response · Authors · 2025-11-28
> > >
> > > Thank you very much! We highly appreciate the insightful suggestions and will update the Pareto Frontier figure in the final version.

---

### Official Review · Reviewer_gGVw · 2025-11-01

**Soundness:** 3
**Presentation:** 3
**Contribution:** 3
**Rating:** 6
**Confidence:** 3

**Summary:**

This paper addresses the degradation in generation quality that with quantization or feature caching. The authors first analyze how approximation errors propagate across diffusion timesteps. Motivated by this, they propose Iterative Error Correction (IEC), a plug-and-play, test-time refinement that injects a fixed-point iteration at selected timesteps. They provide a convergence argument via Banach's fixed-point theorem and show that IEC reduces error growth from exponential to linear in theory. Empirically, IEC consistently improves metrics across multiple datasets and efficiency techniques, with a tunable compute–quality trade-off.

**Strengths:**

- IEC is model-agnostic, requires no retraining, and can be dropped into existing inference pipelines. This plug-and-play nature makes it highly practical for real deployments.
- The method is evaluated across several models and multiple datasets. Improvements are consistent, with especially notable gains for aggressive efficiency settings.
- The paper includes ablations over λ, iteration count K, and which timesteps are refined, and a brief comparison to naïvely adding iterations, helping understand the method better.

**Weaknesses:**

- For Stable Diffusion, IEC is applied only at the first timestep, yielding modest improvements. This raises concerns about the practical benefits at scale if the method cannot be applied more broadly due to compute.
- W4A8 on LSUN-Bedrooms “collapses” without detail. A short analysis of when IEC helps vs. cannot rescue severe degradation would be valuable, potentially guiding users to regimes where IEC is most beneficial.
- Most experiments use T=100 (or 250). Since many efficient systems run at T≤20–50, it would be useful to report results in that low-step regime to demonstrate relevance to contemporary fast samplers.
- Showing ||At + BtJt|| > 1 indicates potential amplification but does not alone establish exponential growth of the end-to-end error. More direct measurement across models/datasets would strengthen the claim.

**Questions:**

Please check Weaknesses

---

> ### Author Response · Authors · 2025-11-20
> **Response to Reviewer gGVw**
>
> Thank you for your insightful advice!
>
> ---
>
> *W1. For Stable Diffusion, IEC is applied only at the first timestep, yielding modest improvements. This raises concerns about the practical benefits at scale if the method cannot be applied more broadly due to compute.*
>
> **Response to W1:**
>
> Applying IEC only at the first timestep for Stable Diffusion was a specific design choice to demonstrate that even minimal intervention (adding negligible overhead) yields metric improvements (e.g., FID improvement from 23.65 to 23.36 in Table 4). However, this is not a limitation of the method. As shown in our ablation study (Fig. 3), IEC exhibits a clear efficiency-quality trade-off, where applying IEC to more timesteps consistently yields better generation quality. Users can flexibly adjust the number of applied steps based on their compute budget. This flexibility allows IEC to be tailored to diverse deployment scenarios, ranging from resource-constrained edge devices (applying IEC only to critical steps) to high-performance servers (applying IEC broadly for maximum fidelity), proving its practical benefits.
>
> ---
>
> *W2. W4A8 on LSUN-Bedrooms “collapses” without detail. A short analysis of when IEC helps vs. cannot rescue severe degradation would be valuable, potentially guiding users to regimes where IEC is most beneficial.*
>
> **Response to W2:**
>
> We appreciate the reviewer's suggestion. We clarify that "collapse" on LSUN-Bedrooms (W4A8) referred to the poor baseline performance, not an inability of IEC to function. For example, for CacheQ-N=10 with W4A8 on LSUN-Bedrooms, the baseline FID was 156.3, indicating severe degradation. While applying IEC drastically improved the FID to 52.6, demonstrating that IEC can significantly recover the signal even from a heavily damaged model.
>
> We agree that analysis of IEC when IEC helps vs. cannot rescue severe degradation is valuable for users. Based on our extensive experiments (including the W4A8 LSUN-Bedrooms case above), we find that IEC remains effective even under severe degradation. We believe IEC is most beneficial when the model preserves the basic denoising direction, and IEC could effectively correct the trajectory deviations to restore high fidelity. However, in cases of total mode collapse (where model prediction is entirely random), IEC cannot recover generation it cannot hallucinate correct features if the guidance signal is entirely lost.
>
> We have added these analyses to the revised paper (Sec A.2 and Sec A.3) to guide users.
>
> ---
>
> *W3. Most experiments use T=100 (or 250). Since many efficient systems run at T≤20–50, it would be useful to report results in that low-step regime to demonstrate relevance to contemporary fast samplers.*
>
> **Response to W3:**
>
> First, we clarify that our Stable Diffusion experiments on MS-COCO were already conducted with T=50 (see Table 4 caption ), which aligns with the reviewer's suggestion. To further demonstrate relevance to contemporary fast samplers and architectures, we added experiments on DiT-XL/2 (ImageNet) with T=20 and T=50. The baseline combines the Q-DiT (W8A8 and W4A8) and Deepcache (N=2 for T=20; N=5 for T=50). The table below shows that IEC significantly improves performance even in these low-step regimes. For instance, on DiT-XL/2 (W8A8, T=20), IEC reduces FID from 7.63 to 6.20 ($\sim$19% improvement), confirming its effectiveness for modern, fast-sampling Transformers.
>
> | Model | W/A | N| FID (smaller is better) | IS (higher is better)|
> | -- | -- | -- |-- |-- |
> | DiT-XL/2 (T=50) | 8/8 | 5 | 6.71 | 193.02|
> | DiT-XL/2 (T=50) + IEC (ours)| 8/8 | 5 | **4.56** | **222.7**|
> |-|
> | DiT-XL/2 (T=50) | 4/8 | 5 | 9.42 | 157.2|
> | DiT-XL/2 (T=50) + IEC (ours)| 4/8 | 5 | **5.53** |**192.8**|
> |-|
> | DiT-XL/2 (T=20) | 8/8 | 2 |7.63|184.2|
> | DiT-XL/2 (T=20) + IEC (ours)| 8/8 | 2 |**6.20**|**198.4**|
> |-|
> | DiT-XL/2 (T=20) | 4/8 | 2 | 10.3|158.4|
> | DiT-XL/2 (T=20) + IEC (ours) | 4/8 | 2 | **9.01**|**167.4**|
>
> ---
>
> *W4. Showing ||At + BtJt|| > 1 indicates potential amplification but does not alone establish exponential growth of the end-to-end error. More direct measurement across models/datasets would strengthen the claim.*
>
>
>
> **Response to W4:**
>
> Thank you! We have also provided a direct empirical error comparison in the paper. As shown in Figure 2, the accumulated error magnitude over time for CIFAR-10 shows a clear, rapid rise. We have added measurements on (LSUN-Church and Bedrooms, shown in the table below. The error magnitude explodes from $<0.1$ at $T=1$ to $>140$ (or even $>400$) at $T=100$. This rapid amplification empirically confirms the exponential error propagation predicted by our theory.
>
> | LSUN-Churchs |T=1|T=20| T=40| T=60| T=80| T=100|
> | -- | -- | -- |-- |-- |--|--|
> |W8A8 |0.007|0.205|0.592|2.72|30.22|141.5|
> |Deepcache (N=10) |5.89e-06|0.15|2.51| 14.69|64.89|188.6|
> |-|
> | LSUN- Bedrooms|T=1 | T=20| T=40| T=60| T=80| T=100|
> |W8A8 |0.04|1.35|7.12|38.06|143.8|428.1|
> |Deepcache (N=10) |1.65e-05|0.10|2.88|40.00|204.7|598.7|

---

### Comment · Area_Chair_PHqv · 2025-11-21

Dear Reviewers,

We kindly encourage you to review and respond to the authors’ rebuttals. Your timely feedback is important for ensuring a fair and thorough review process. Thank you for your contributions to ICLR 2026.

AC

---

### Author Response · Authors · 2025-12-02
**Summary Comment**

**Dear Area Chair**,

We are grateful for your efforts and acknowledge the heavy load caused by the recent data leakage (**around Nov 27, 2025, 03:00, AoE**). Since the system freeze restricted reviewers from updating scores/responding, we offer this brief summary to assist your evaluation. It outlines the confirmed score increase (before data leakage) and the key revisions made in direct response to reviewer comments.

**Summary of our paper:** To address the issue where approximation errors in efficient diffusion models accumulate exponentially across timesteps, we propose Iterative Error Correction (IEC), a novel test-time method designed to mitigate these errors without requiring retraining or architectural changes. By iteratively refining model outputs, IEC is theoretically proven to reduce error propagation growth from exponential to linear, thereby serving as a flexible and generalizable solution that consistently enhances generation quality across diverse architectures and efficiency settings.

---

# Summary of Reviewer Status
Current Official Avg Rating: 5.5 → Revised Avg Rating before data leakage: 6.0
|Reviewer|Score|Reviewer Response |Key Comments|
|--|--|--|--|
|gGVw|6|No| 'highly practical for real deployments', 'Improvements are consistent', 'notable gains' |
|NVsH|6|Yes| 'It targets the practical, post-deployment scenario for efficient models, which is somewhat under-explored', 'The theoretical analysis is of good quality'|
|cbcn|4 -> 6|Yes| Explicitly raised score on **Nov 26, 2025, 04:49 AoE** (at least 22 hours before data leakage).  'I have updated my rating.'|
|roVY|6 |No| 'highly practical', 'theoretical grounding strengthens the credibility and generality of the proposed method', 'particularly valuable for real-world scenarios'|

---

## 1. Reviewer gGVw: Positive (Score: 6)
**Initial Concern:** The reviewer initially questions the boundaries of our IEC method in other settings (Stable Diffusion, low sampling steps, and W4A8 cases) and requests direct experimental evidence to substantiate the theoretical claims of error amplification.

**Rebuttal:** We addressed the reviewer's concerns by clarifying the flexible efficiency-quality trade-off of IEC, providing additional experiments in low-step (T=20/50), adding results on W4A8 cases and related clarification, and offering empirical error data to confirm the theoretical prediction of exponential error amplification.

---

## 2. Reviewer NVsH: Positive (Score: 6)
**Initial Concern:** The reviewer suggests applying our IEC to other under-trained diffusion models, requests comprehensive Pareto frontier plots to analyze compute-performance trade-offs, asks for validation on flow-based models, and highlights presentation issues regarding table redundancy and missing data.

**Rebuttal:** We provide results on under-trained diffusion models to clarify IEC's specific design for efficiency errors, provided Pareto plots to demonstrate superior trade-offs, validated the method on flow-based models, and improved the presentation by merging tables and including missing data.

**Outcome:** The reviewer explicitly stated (at Nov 27, 2025, 02:49 AoE):

> I thank the authors for their detailed responses and the additional experiments. The authors have successfully addressed most of my concerns, and I am inclined to recommend acceptance.

---

## 3. Reviewer cbcn: Positive Score Raise (4 -> 6)
**Initial Concern:** The reviewer suggests clarifying the setting and the fairness of the comparison, requesting a baseline with a more compute budget and a proof of exponential error growth, and a discussion about the comparison with DPM-Solver++, consistency models, and other iterative refinement approaches.

**Rebuttal:** We have addressed these concerns by clarifying the main target of our IEC, introducing a baseline with more compute budget, providing both theoretical derivation and empirical data to verify exponential error growth, and clarifying IEC's distinction from DPM-Solver++ and consistency models.

**Outcome:** The reviewer explicitly stated (Nov 26, 2025, 04:49 AoE):

> Thank you for providing additional results and explanations. Most of my concerns are addressed in the rebuttal. I have updated my rating.

---

## 4. Reviewer roVY: Positive (Score: 6)

**Initial Concern:**  The reviewer suggests requesting validation on transformer-based architectures and non-DDIM samplers (e.g., DPM-Solver), inquiring about its applicability to adaptive computation methods, and asking for missing implementation details and discussion about the applicability of IEC on other acceleration methods.

**Rebuttal:** We addressed the generalizability concerns by conducting additional experiments on transformer-based architectures (DiT), clarifying the theoretical compatibility with non-DDIM samplers (DPM-Solver) and adaptive computation methods, and providing the missing implementation details and required discussion.

---

### Meta-Review · Area_Chair_4p3S · 2026-01-05

**Summary:**

This paper receives 1 negative review (4), and 3 positive reviews (6) initially. Reviewer cbcn increased his/her score after the rebuttal and discussion period. Initial concerns mainly lie in the comprehensiveness of experiments, such as limited to CNN-based backbone, comparison to baselines is unfair, behaviors in low-step regime, missing analysis of some experiment results, etc. During the rebuttal and discussion period, authors successfully provided corresponding results and analysis, resolving these concerns. AC found no further concerns, thus the decision is accept.

**Reviewer Concerns:**

Concerns are mainly focused on the comprehensiveness of experiments, and these concerns are well addressed by authors' responses.

**Reviewer Scores:**

All reviewers will stay positive.

---

### Decision · Program_Chairs · 2026-01-26

Accept (Poster)